# InstaHide's Sample Complexity When Mixing Two Private Images

## Abstract

Inspired by InstaHide challenge [Huang, Song, Li and Arora'20], [Chen, Song and Zhuo'20] recently provides one mathematical formulation of InstaHide attack problem under Gaussian images distribution. They show that it suffices to use $O(n_{\mathsf{priv}}^{k_{\mathsf{priv}}-2/(k_{\mathsf{priv}}+1)})$ samples to recover one private image in $n_{\mathsf{priv}}^{O(k_{\mathsf{priv}})} + \mathrm{poly}(n_{\mathsf{pub}})$ time for any integer $k_{\mathsf{priv}}$, where $n_{\mathsf{priv}}$ and $n_{\mathsf{pub}}$ denote the number of images used in the private and the public dataset to generate a mixed image sample. Under the current setup for the InstaHide challenge of mixing two private images ($k_{\mathsf{priv}} = 2$), this means $n_{\mathsf{priv}}^{4/3}$ samples are sufficient to recover a private image. In this work, we show that $n_{\mathsf{priv}} \log(n_{\mathsf{priv}})$ samples are sufficient (information-theoretically) for recovering all the private images.

## 1 Introduction

Collaboratively training neural networks based on sensitive data is appealing in many AI applications, such as healthcare, fraud detection, and virtual assistants. How to train neural networks without compromising data confidentiality and prediction accuracy has become an important and common research goal Shokri & Shmatikov (2015); Ryffel et al. (2018); Phong et al. (2018); McMahan et al. (2017); Konečnỳ et al. (2016) in both academia and industry.

Huang et al. (2020b) recently proposed an approach called InstaHide for image classification. The key idea is to train the model on a dataset where (1) each image is a mix of $k_{\mathsf{priv}}$ private images and $k_{\mathsf{pub}}$ public images, and (2) each pixel is independently sign-flipped after the mixing. Instahide shows promising prediction accuracy on MNIST Deng (2012), CIFAR-10 Krizhevsky (2012), CIFAR-100, and ImageNet datasets Deng et al. (2009). TextHide Huang et al. (2020a) applies InstaHide's idea to text datasets and achieves promising results on natural language processing tasks.

To understand the security aspect of InstaHide in realistic deployment scenarios, InstaHide authors present an InstaHide challenge Challenge (2020) that involves $n_{\mathsf{priv}} = 100$ private images, ImageNet dataset as the public images, $m = 5000$ sample images (each image is a combination of $k_{\mathsf{priv}} = 2$ private images and $k_{\mathsf{pub}} = 4$ public images and the sign of each pixel is randomly flipped). The challenge is to recover a private image given the set of sample images.

Chen et al. (2021) is a theoretical work that formulates the InstaHide attack problem as a recovery problem. It also provides an algorithm to recover a private image assuming each private and public image is a random Gaussian image (i.e., each pixel is an i.i.d. draw from $\mathcal{N}(0, 1)$). The algorithm shows that $O(n_{\mathsf{priv}}^{k_{\mathsf{priv}}-2/(k_{\mathsf{priv}}+1)})$ sample images are sufficient to recover one private image. Carlini et al. (2020) provides the first heuristic-based practical attack for the InstaHide challenge ($k_{\mathsf{priv}} = 2$), and it can generate images that are visually similar to the private images in the InstaHide challenge dataset. Luo et al. (2021) provides the first heuristic-based practical attack for the InstaHide challenge ($k_{\mathsf{priv}} = 2$) when data augmentation is enabled.

Although the InstaHide challenge is considered broken by several researchers, the current InstaHide challenge is itself too simple, and it is unclear whether the existing attacks Carlini

et al. (2020); Luo et al. (2021) can still work when we use InstaHide to protect large numbers of private images (large $n$) Arora (2020).

This raises an important question:

*What's the minimal number of InstaHide images needed to recover a private image?*

This question is worthwhile to consider because it is a quantitative measure for how secure InstaHide is. With the same formulation in Chen et al. (2021), we achieve a better upper bound on the number of samples needed to recover private images when $k_{\mathsf{priv}} = 2$. Our new algorithm can recover all the private images using only $\Omega(n_{\mathsf{priv}} \log(n_{\mathsf{priv}}))$ samples.[1] This significantly improves the state-of-the-art theoretical results Chen et al. (2021) that requires $n_{\mathsf{priv}}^{4/3}$ samples to recover a single private image. However, our running time is exponential in the number of private images ($n_{\mathsf{priv}}$) and polynomial in the number of public images ($n_{\mathsf{pub}}$), where the running time of the algorithm in Chen et al. (2021) is polynomial in $n_{\mathsf{priv}}$ and $n_{\mathsf{pub}}$. In addition, we provide a four-step framework to compare our attacks with the attacks presented in Carlini et al. (2020) and Chen et al. (2021). We hope our framework can inspire more efficient attacks on InstaHide-like approaches and can guide the design of future-generation deep learning algorithms on sensitive data.

## 1.1 Our result

Chen et al. (2021) formulates the InstaHide attack problem as a recovery problem that given sample access to an oracle that can generate as much as InstaHide images you want, there are two goals : 1) sample complexity, minimize the number InstaHide images being used, 2) running time, use those InstaHide images to recover the original images as fast as possible.

Similar to Chen et al. (2021), we consider the case where private and public data vectors are Gaussians. Let $S_{\mathsf{pub}}$ denote the set of public images with $|S_{\mathsf{pub}}| = n_{\mathsf{pub}}$, let $S_{\mathsf{priv}}$ denote the set of private images with $|S_{\mathsf{priv}}| = n_{\mathsf{priv}}$. The model that produces InstaHide image can be described as follows:

- Pick $k_{\mathsf{pub}}$ vectors from public data set and $k_{\mathsf{priv}}$ vectors from private data set.
- Normalize $k_{\mathsf{pub}}$ vectors by $1/\sqrt{k_{\mathsf{pub}}}$ and normalize $k_{\mathsf{priv}}$ vectors by $1/\sqrt{k_{\mathsf{priv}}}$.
- Add $k_{\mathsf{pub}} + k_{\mathsf{priv}}$ vectors together to obtain a new vector, then flip each coordinate of that new vector independently with probability $1/2$.

We state our result as follows:

**Theorem 1.1** (Informal version of Theorem 3.1)**.** *Let $k_{\mathsf{priv}} = 2$. If there are $n_{\mathsf{priv}}$ private vectors and $n_{\mathsf{pub}}$ public vectors, each of which is an i.i.d. draw from $\mathcal{N}(0, \mathsf{Id}_d)$, then as long as*

$$d = \Omega(\operatorname{poly}(k_{\mathsf{pub}}) \log(n_{\mathsf{pub}} + n_{\mathsf{priv}})),$$

*there is some $m = O(n_{\mathsf{priv}} \log n_{\mathsf{priv}})$ such that, given a sample of $m$ random synthetic vectors independently generated as above, one can exactly recover all the private vectors in time*

$$O(dm^2 + dn_{\mathsf{pub}}^2 + n_{\mathsf{pub}}^{2\omega+1} + mn_{\mathsf{pub}}^2) + d2^{O(m)}$$

*with high probability.*

**Notations.** For any positive integer $n$, we use $[n]$ to denote the set $\{1, 2, \cdots, n\}$. For a set $S$, we use $\operatorname{supp}(S)$ to denote the support of $S$, i.e., the indices of its elements. We also use $\operatorname{supp}(w)$ to denote the support of vector $w \in \mathbb{R}^n$, i.e. the indices of its non-zero coordinates. For a vector $x$, we use $\|x\|_2$ to denotes entry-wise $\ell_2$ norm. For two vectors $a$ and $b$, we use $a \circ b$ to denote a vector where $i$-th entry is $a_i b_i$. For a vector $a$, we use $|a|$ to denote a vector where the $i$-th entry is $|a_i|$. Given a vector $v \in \mathbb{R}^n$ and a subset $S \subset [n]$ we use $[v]_S \in \mathbb{R}^{|S|}$ to denote the restriction of $v$ to the coordinates indexed by $S$.

---

[1]For the worst case distribution, $\Omega(n_{\mathsf{priv}})$ is a trivial sample complexity lower bound.

**Contributions.** Our contributions can be summarized into the following folds.

- We propose an algorithm that recover all the private images using only $\Omega(n_{\mathsf{priv}} \log(n_{\mathsf{priv}}))$ samples in the recent theoretical framework of attacking InstaHide Huang et al. (2020b) when mixing two private image, improving the state-of-art result of Chen et al. (2021).

- We summarize existing methods of attacking InstaHide into a unifying framework. By examining the functionality of each steps we identify the connection of a key step with problems in graph isomorphism. We also reveal the vulnerability of existing method to recover all private images by showing hardness to recover all images.

**Organizations.** In Section 2 we formulate our attack problem. In Section 3 we present our algorithm and main results. In Section 4 we conclude our paper and discuss future directions.

## 2 PRELIMINARIES

We use the same setup as Chen et al. (2021).

**Definition 2.1** (Image matrix notation, Definition 2.2 in Chen et al. (2021)). *Let image matrix $\mathbf{X} \in \mathbb{R}^{d \times n}$ be a matrix whose columns consist of vectors $x_1, \ldots, x_n \in \mathbb{R}^d$ corresponding to $n$ images each with $d$ pixels taking values in $\mathbb{R}$. It will also be convenient to refer to the rows of $\mathbf{X}$ as $p_1, \ldots, p_d \in \mathbb{R}^n$.*

We define public set, private set and synthetic images following the setup in Huang et al. (2020b).

**Definition 2.2** (Public/private notation, Definition 2.3 in Chen et al. (2021)). *We will refer to $S_{\mathsf{pub}} \subset [n]$ and $S_{\mathsf{priv}} = [n] \backslash S_{\mathsf{pub}}$ as the set of public and private images respectively, and given a vector $w \in \mathbb{R}^n$, we will refer to $\operatorname{supp}(w) \cap S_{\mathsf{pub}}$ and $\operatorname{supp}(w) \cap S_{\mathsf{priv}}$ as the public and private coordinates of $w$ respectively.*

**Definition 2.3** (Synthetic images, Definition 2.4 in Chen et al. (2021)). *Given sparsity levels $k_{\mathsf{pub}} \leq |S_{\mathsf{pub}}|, k_{\mathsf{priv}} \leq |S_{\mathsf{priv}}|$, image matrix $\mathbf{X} \in \mathbb{R}^{d \times n}$ and a selection vector $w \in \mathbb{R}^n$ for which $[w]_{S_{\mathsf{pub}}}$ and $[w]_{S_{\mathsf{priv}}}$ are $k_{\mathsf{pub}}$- and $k_{\mathsf{priv}}$-sparse respectively, the corresponding synthetic image is the vector $y^{\mathbf{X},w} = |\mathbf{X}w| \in \mathbb{R}^d$ where $|\cdot|$ denotes entrywise absolute value. We say that $\mathbf{X} \in \mathbb{R}^{d \times n}$ and a sequence of selection vectors $w_1, \ldots, w_m \in \mathbb{R}^n$ give rise to a synthetic dataset $\mathbf{Y} \in \mathbb{R}^{m \times d}$ consisting of the images $(y^{\mathbf{X},w_1}, \ldots, y^{\mathbf{X},w_m})^\top$.*

We consider Gaussian image which is a common setting in phase retrieval Candes et al. (2013); Netrapalli et al. (2017); Candes et al. (2015).

**Definition 2.4** (Gaussian images, Definition 2.5 in Chen et al. (2021)). *We say that $\mathbf{X}$ is a random Gaussian image matrix if its entries are sampled i.i.d. from $\mathcal{N}(0,1)$.*

Distribution over selection vectors follows from variants of Mixup Zhang et al. (2017). Here we $\ell_2$ normalize all vectors for convenience of analysis. Since $k_{\mathsf{priv}}$ is a small constant, our analysis can be easily generalized to other normalizations.

**Definition 2.5** (Distribution over selection vectors, Definition 2.6 in Chen et al. (2021)). *Let $\mathcal{D}$ be the distribution over selection vectors defined as follows. To sample once from $\mathcal{D}$, draw random subset $T_1 \subset S_{\mathsf{pub}}, T_2 \subseteq S_{\mathsf{priv}}$ of size $k_{\mathsf{pub}}$ and $k_{\mathsf{priv}}$ and output the unit vector whose $i$-th entry is $1/\sqrt{k_{\mathsf{pub}}}$ if $i \in T_1$, $1/\sqrt{k_{\mathsf{priv}}}$ if $i \in T_2$, and zero otherwise.[2]*

For convenience we will define pub and priv operators as follows,

**Definition 2.6** (Public/private operators). *We define function $\mathsf{pub}(\cdot)$ and $\mathsf{priv}(\cdot)$ such that for vector $w \in \mathbb{R}^n$, $\mathsf{pub}(w) \in \mathbb{R}^{n_{\mathsf{pub}}}$ will be the vector which only contains the coordinates of $w$ corresponding to the public subset $S_{\mathsf{pub}}$, and $\mathsf{priv}(w) \in \mathbb{R}^{n_{\mathsf{priv}}}$ will be the vector which only contains the coordinates of $w$ corresponding to the private subset $S_{\mathsf{priv}}$.*

---

[2]Note that any such vector does not specify a convex combination, but this choice of normalization is just to make some of the analysis later on somewhat cleaner, and our results would still hold if we chose the vectors in the support of $\mathcal{D}$ to have entries summing to 1.

---

**Algorithm 1** Recovering All Private Images when $k_{\mathsf{priv}} = 2$

---

1: **procedure** RECOVERALL($\mathbf{Y}$)                                            ▷ Theorem 3.1, Theorem 1.1
2:                                            ▷ InstaHide dataset $\mathbf{Y} = (y^{\mathbf{X}, w_1}, \ldots, y^{\mathbf{X}, w_m})^\top \in \mathbb{R}^{m \times d}$
3:                                            ▷ Step 1. Retrieve Gram matrix
4:     $\mathbf{M} \leftarrow \frac{1}{k_{\mathsf{priv}} + k_{\mathsf{pub}}} \cdot$ GRAMEXTRACT($\mathbf{Y}, \frac{1}{2(k_{\mathsf{pub}} + k_{\mathsf{priv}})}$)        ▷ Algorithm 1 in Chen et al. (2021)
5:                                            ▷ Step 2. Subtract Public images from Gram matrix
6:     **for** $i \in [m]$ **do**
7:         $S_i \leftarrow$ LEARNPUBLIC($\{((p_j)_{S_{\mathsf{pub}}}, y_j^{\mathbf{X}, w_i})\}_{j \in [d]}$)        ▷ Algorithm 2 in Chen et al. (2021)
8:     **end for**
9:     $\mathbf{W}_{\mathsf{pub}} \leftarrow (\mathsf{pub}(\mathrm{vec}(S_1)), \ldots, \mathsf{pub}(\mathrm{vec}(S_m)))^\top$                ▷ $\mathbf{W}_{\mathsf{pub}} \in \{0, 1\}^{m \times n_{\mathsf{pub}}}$
10:    $\mathbf{M}_{\mathsf{priv}} \leftarrow k_{\mathsf{priv}} \cdot (\mathbf{M} - \frac{1}{k_{\mathsf{pub}}} \mathbf{W}_{\mathsf{pub}} \mathbf{W}_{\mathsf{pub}}^\top)$
11:                                            ▷ Step 3. Assign original images
12:    $\mathbf{W}_{\mathsf{priv}} \leftarrow$ ASSIGNINGORIGINALIMAGES($\mathbf{M}_{\mathsf{priv}}, n_{\mathsf{priv}}$)                ▷ Algorithm 2
13:                                            ▷ Step 4. Solving system of equations.
14:    $\mathbf{Y}_{\mathsf{pub}} = \frac{1}{\sqrt{k_{\mathsf{pub}}}} \mathbf{W}_{\mathsf{pub}} \mathbf{X}_{\mathsf{pub}}^\top$        ▷ $\mathbf{X}_{\mathsf{pub}} \in \mathbb{R}^{d \times n_{\mathsf{pub}}}, \mathbf{Y}_{\mathsf{pub}} \in \mathbb{R}^{m \times d}, \mathbf{W}_{\mathsf{pub}} \in \{0,1\}^{m \times n_{\mathsf{pub}}}$
15:    $\widetilde{X} \leftarrow$ SOLVINGSYSTEMOFEQUATIONS($\mathbf{W}_{\mathsf{priv}}, \sqrt{k_{\mathsf{priv}}} \mathbf{Y}_{\mathsf{pub}}, \sqrt{k_{\mathsf{priv}}} \mathbf{Y}$)                ▷ Algorithm 3
16:    **return** $\widetilde{X}$
17: **end procedure**

---

For subset $\widetilde{S} \subset S$ we will refer to $\mathrm{vec}(\widetilde{S}) \in \mathbb{R}^n$ as the vector that $\mathrm{vec}(\widetilde{S})_i = 1$ if $i \in \widetilde{S}$ and $\mathrm{vec}(\widetilde{S})_i = 0$ otherwise. We define the public and private components of $\mathbf{W}$ and $\mathbf{Y}$ for convenience.

**Definition 2.7** (Public and private components of image matrix and selection vectors). *For a sequence of selection vectors $w_1, \ldots, w_m \in \mathbb{R}^n$ we will refer to*

$$\mathbf{W} = (w_1, \ldots, w_m)^\top \in \mathbb{R}^{m \times n}$$

*as the mixup matrix.*

*Specifically, we will refer to $\mathbf{W}_{\mathsf{pub}} \in \{0, 1\}^{m \times n_{\mathsf{pub}}}$ as the public component of mixup matrix and $\mathbf{W}_{\mathsf{priv}} \in \{0, 1\}^{m \times n_{\mathsf{priv}}}$ as the private component of mixup matrix, i.e.,*

$$\mathbf{W}_{\mathsf{pub}} = \sqrt{k_{\mathsf{pub}}} \cdot \begin{bmatrix} \mathsf{pub}(\mathbf{W}_{1,*}) \\ \vdots \\ \mathsf{pub}(\mathbf{W}_{m,*}) \end{bmatrix} \in \{0, 1\}^{m \times n_{\mathsf{pub}}}, \mathbf{W}_{\mathsf{priv}} = \sqrt{k_{\mathsf{priv}}} \cdot \begin{bmatrix} \mathsf{priv}(\mathbf{W}_{1,*}) \\ \vdots \\ \mathsf{priv}(\mathbf{W}_{m,*}) \end{bmatrix} \in \{0, 1\}^{m \times n_{\mathsf{priv}}}.$$

*We will refer to $\mathbf{X}_{\mathsf{pub}} \in \mathbb{R}^{d \times n_{\mathsf{pub}}}$ as public component of image matrix which only contains the columns of $\mathbf{X} \in \mathbb{R}^{d \times n}$ corresponding to the public subset $S_{\mathsf{pub}}$, and $\mathbf{X}_{\mathsf{priv}} \in \mathbb{R}^{d \times n_{\mathsf{priv}}}$ as private component of image matrix which only contains the columns of $\mathbf{X} \in \mathbb{R}^{d \times n}$ corresponding to the private subset $S_{\mathsf{priv}}$.*

*Furthermore we define $\mathbf{Y}_{\mathsf{pub}} \in \mathbb{R}^{m \times d}$ as public contribution to InstaHide images and $\mathbf{Y}_{\mathsf{priv}} \in \mathbb{R}^{m \times d}$ as private contribution to InstaHide images:*

$$\mathbf{Y}_{\mathsf{pub}} = \frac{1}{\sqrt{k_{\mathsf{pub}}}} \mathbf{W}_{\mathsf{pub}} \mathbf{X}_{\mathsf{pub}}^\top, \quad \mathbf{Y}_{\mathsf{priv}} = \frac{1}{\sqrt{k_{\mathsf{priv}}}} \mathbf{W}_{\mathsf{priv}} \mathbf{X}_{\mathsf{priv}}^\top.$$

Instead of considering only one private image recovery as Chen et al. (2021), here we consider a harder question which requires to recover all the private images.

---

**Problem 1** (Exact Private image recovery). *Let $\mathbf{X} \in \mathbb{R}^{d \times n}$ be a Gaussian image matrix. Given access to the public images $\{x_s\}_{s \in S_{\mathsf{pub}}}$ and the synthetic dataset $(y^{\mathbf{X}, w_1}, \ldots, y^{\mathbf{X}, w_m})$, where $w_1, \ldots, w_m \sim \mathcal{D}$ are unknown selection vectors, output a set of vectors $\{\widetilde{x}_s\}_{s \in S_{\mathsf{priv}}}$ for which there exists a one-to-one mapping $\phi$ from $\{\widetilde{x}_s\}_{s \in S_{\mathsf{priv}}}$ to $\{x_s\}_{s \in S_{\mathsf{priv}}}$ satisfying $\phi(\widetilde{x}_s)_j = (x_s)_j, \forall j \in [d]$.*

---

## 3    Recovering All Private Images when $k_{\mathsf{priv}} = 2$

In this section, we prove our main algorithmic result. Our algorithm follows the high-level procedure introduced in section A. The details are elaborated in following subsections.

**Theorem 3.1** (Main result). *Let $S_{\mathsf{pub}} \subset [n]$, and let $n_{\mathsf{pub}} = |S_{\mathsf{pub}}|$ and $n_{\mathsf{priv}} = |S_{\mathsf{priv}}|$. Let $k_{\mathsf{priv}} = 2$. Let $k = k_{\mathsf{priv}} + k_{\mathsf{pub}}$. If $d \geq \Omega\big(\operatorname{poly}(k_{\mathsf{pub}}, k_{\mathsf{priv}}) \log(n_{\mathsf{pub}} + n_{\mathsf{priv}})\big)$ and $m \geq \Omega\big(k^{\operatorname{poly}(k_{\mathsf{priv}})} n_{\mathsf{priv}} \log n_{\mathsf{priv}}\big)$, then with high probability over $\mathbf{X}$ and the sequence of randomly chosen selection vectors $w_1, \ldots, w_m \sim \mathcal{D}$, there is an algorithm which takes as input the synthetic dataset $\mathbf{Y}^\top = (y^{\mathbf{X}, w_1}, \ldots, y^{\mathbf{X}, w_m}) \in \mathbb{R}^{d \times m}$ and the columns of $\mathbf{X}$ indexed by $S_{\mathsf{pub}}$, and outputs $n_{\mathsf{priv}}$ images $\{\widetilde{x}_s\}_{s \in S_{\mathsf{priv}}}$ for which there exists one-to-one mapping $\phi$ from $\{\widetilde{x}_s\}_{s \in S_{\mathsf{priv}}}$ to $\{x_s\}_{s \in S_{\mathsf{priv}}}$ satisfying $\phi(\widetilde{x}_s)_j = (x_s)_j$ for all $j \in [d]$. Furthermore, the algorithm runs in time*

$$O(m^2 d + d n_{\mathsf{pub}}^2 + n_{\mathsf{pub}}^{2\omega+1} + m n_{\mathsf{priv}}^2 + 2^m \cdot m n_{\mathsf{priv}}^2 d).$$

**Remark 3.2.** *Our result improves on Chen et al. (2021) on two aspects. First, we reduce the sample complexity from $n_{\mathsf{priv}}^{k_{\mathsf{priv}} - 2/(k_{\mathsf{priv}}+1)}$ to $n_{\mathsf{priv}} \log n_{\mathsf{priv}}$ when $k_{\mathsf{priv}} = 2$. Note that our sample complexity is optimal up to logarithmic factors since finding unique solutions of linear system requires at least $n_{\mathsf{priv}}$ sample complexity. Second, we can recover all private images exactly rather than recovering a single image, which is highly desirable for real-world practitioners. Furthermore notice that fixing all public images, multiplying any private image by $-1$ might not keep* InstaHide *images unchanged. Thus information theoretically, we are able to recover all private images precisely (not only absolute values) as long as we have access to sufficient synthetic images. In fact, from the proof of Lemma 3.7 our sample complexity suffices to achieve exact recovery.*

### 3.1    Retrieving Gram matrix

In this section, we present the algorithm for retrieving the Gram matrix.

**Lemma 3.3** (Retrieve Gram matrix, Chen et al. (2021)). *Let $n = n_{\mathsf{pub}} + n_{\mathsf{priv}}$. Suppose $d = \Omega(\log(m/\delta)/\eta^4)$. For a random Gaussian image matrix $\mathbf{X} \in \mathbb{R}^{d \times n}$ and arbitrary $w_1, \ldots, w_m \in \mathbb{S}_{\geq 0}^{d-1}$, let $\Sigma^*$ be the output of $\mathrm{GRAMEXTRACT}$ when we set $\eta = 1/2k$. Then with probability $1 - \delta$ over the randomness of $\mathbf{X}$, we have that $\Sigma^* = k \cdot \mathbf{W}\mathbf{W}^\top \in \mathbb{R}^{m \times m}$. Furthermore, $\mathrm{GRAMEXTRACT}$ runs in time $O(m^2 d)$.*

We briefly describe how this is achieved. Without loss of generality, we may assume $S_{\mathsf{priv}} = [n]$, since once we determine the support of public images $S_{\mathsf{pub}}$, we can easily subtract the contribution of them. Consider a matrix $\mathbf{Y} \in \mathbb{R}^{m \times d}$ whose rows are $y^{\mathbf{X}, w_1}, \ldots, y^{\mathbf{X}, w_m}$. Then, it can be written as

$$\mathbf{Y} = \begin{bmatrix} |\langle p_1, w_1 \rangle| & \cdots & |\langle p_d, w_1 \rangle| \\ \vdots & \ddots & \vdots \\ |\langle p_1, w_m \rangle| & \cdots & |\langle p_d, w_m \rangle|. \end{bmatrix}$$

Since $\mathbf{X}$ is a Gaussian matrix, we can see that each column of $\mathbf{Y}$ is the absolute value of an independent draw of $\mathcal{N}(0, \mathbf{W}\mathbf{W}^\top)$. We define this distribution as $\mathcal{N}^{\mathsf{fold}}(0, \mathbf{W}\mathbf{W}^\top)$, and it can be proved that the covariance matrix of $\mathcal{N}^{\mathsf{fold}}(0, \mathbf{W}\mathbf{W}^\top)$ can be directly related $\mathbf{W}\mathbf{W}^\top$. Then, the task becomes estimating the covariance matrix of $\mathcal{N}^{\mathsf{fold}}(0, \mathbf{W}\mathbf{W}^\top)$ from $d$ independent samples (columns of $\mathbf{Y}$), which can be done by computing the empirical estimates.

### 3.2    Remove public images

In this section, we present the algorithm of subtracting public images from Gram matrix. Formally, given any synthetic image $y^{\mathbf{X}, w}$ we recover the entire support of $[w]_{S_{\mathsf{pub}}}$ (essentially $\operatorname{supp}([w]_{S_{\mathsf{pub}}})$).

**Lemma 3.4** (Subtract public images from Gram matrix, Chen et al. (2021)). *Let $n = n_{\mathsf{priv}} + n_{\mathsf{pub}}$. For any $\delta \geq 0$, if $d = \Omega(\mathrm{poly}(k_{\mathsf{pub}})/\log(n/\delta))$, then with probability at least $1 - \delta$ over the randomness of $\mathbf{X}$, we have that the coordinates output by LEARNPUBLIC are exactly equal to $\mathrm{supp}([w]_{S_{\mathsf{pub}}})$. Furthermore, LEARNPUBLIC runs in time $O(dn_{\mathsf{pub}}^2 + n_{\mathsf{pub}}^{2\omega+1})$, where $\omega \approx 2.373$ is the exponent of matrix multiplication Williams (2012).*

Note that this problem is closely related to the Gaussian phase retrieval problem. However, we can only access the public subset of coordinates for any image vector $p_i$. We denote these partial vectors as $\{[p_i]_{S_{\mathsf{pub}}}\}_{i\in[d]}$. The first step is to construct a matrix $\widetilde{\mathbf{M}} \in \mathbb{R}^{n_{\mathsf{pub}} \times n_{\mathsf{pub}}}$:

$$\widetilde{\mathbf{M}} = \frac{1}{d} \sum_{i=1}^{d} ((y_i^{\mathbf{X},w})^2 - 1) \cdot ([p_i]_{S_{\mathsf{pub}}} [p_i]_{S_{\mathsf{pub}}}^{\top} - \mathbf{I}).$$

It can be proved that when $p_i$'s are i.i.d standard Gaussian vectors, the expectation of $\widetilde{\mathbf{M}}$ is $\mathbf{M} = \frac{1}{2}[w]_{S_{\mathsf{pub}}}[w]_{S_{\mathsf{pub}}}^{\top}$. However, when $d \ll n$, $\widetilde{\mathbf{M}}$ is not a sufficiently good spectral approximation of $\mathbf{M}$, which means we cannot directly use the top eigenvector of $\widetilde{\mathbf{M}}$. Instead, with high probability $[w]_{S_{\mathsf{pub}}}$ can be approximated by the top eigenvector of the solution of the following semi-definite programming (SDP):

$$\max_{Z \succeq 0} \langle \widetilde{\mathbf{M}}, Z \rangle \ s.t. \ \mathrm{tr}[Z] = 1, \sum_{i,j=1}^{n_{\mathsf{pub}}} |Z_{i,j}| \leq k_{\mathsf{pub}}.$$

Hence, the time complexity of this step is $O(dn_{\mathsf{pub}}^2 + n_{\mathsf{pub}}^{2\omega+1})$, where the first term is the time cost for constructing $\widetilde{\mathbf{M}}$ and the second term is the time cost for SDP Jiang et al. (2020).

### 3.3 Assigning encoded images to original images

We are now at the position of recovering $\mathbf{W}_{\mathsf{priv}} \in \mathbb{R}^{m \times n_{\mathsf{priv}}}$ from private Gram matrix $\mathbf{M}_{\mathsf{priv}} \in \mathbb{R}^{m \times m}$. Recall that $\mathbf{M}_{\mathsf{priv}} = \mathbf{W}_{\mathsf{priv}} \mathbf{W}_{\mathsf{priv}}^{\top} \in \mathbb{R}^{m \times m}$ where $\mathbf{W}_{\mathsf{priv}} \in \{0,1\}^{m \times n_{\mathsf{priv}}}$ is the mixup matrix with column sparsity $k_{\mathsf{priv}}$. By recovering mixup matrix $\mathbf{W}$ from private Gram matrix $\mathbf{M}$ the attacker maps each synthetic image $y^{\mathbf{X},w_i}, i \in [m]$ to two original images $x_{i_1}, \ldots, x_{i_{k_{\mathsf{priv}}}}$ (to be recovered in the next step) in the private data set, where $k_{\mathsf{priv}} = 2$.

On the other hand, in order to recover original image $x_i$ from private data set, the attacker needs to know precisely the set of synthetic images $y^{\mathbf{X},w_i}, i \in [m]$ generated by $x_i$. Therefore this step is crucial to recover the original private images from InstaHide images. We provide an algorithm and certify that it outputs the private component of mixup matrix with sample complexity $m = \Omega(n_{\mathsf{priv}} \log n_{\mathsf{priv}})$.

As noted by Chen et al. (2021), the intricacy of this step lies in the fact that a family of sets may not be uniquely identifiable from the cardinality of all pairwise intersections. This problem is formally stated in the following.

---

**Problem 2** (Recover sets from cardinality of pairwise intersections). *Let $S_i \subset [n], i \in [m]$ be $n$ sets with cardinality $k$. Given access to the cardinality of pairwise intersections $|S_i \cap S_j|$ for all $i, j \in [m]$, output a family of sets $\widetilde{S}_i \subset [n], i \in [m]$ for which there exists a one-to-one mapping $\phi$ from $\widetilde{S}_i, i \in [m]$ to $S_i, i \in [m]$ satisfying $\phi(\widetilde{S}_j) = S_j$ for all $j \in [m]$.*

---

In real world applications, attackers may not even have access to precise cardinality of pairwise intersections $|S_i \cap S_j|$ for all $i, j \in [m]$ due to errors in retrieving Gram matrix and public coordinates. Instead, attackers often face a harder version of the above problem, where they only know whether $|S_i \cap S_j|$ is an empty set for $i, j \in [m]$. However for mixing two private images, the two problems are the same.

We now provide a solution to this problem. First we define a concept closely related to the above problem.

---

**Algorithm 2** Assigning Original Images

---

1: **procedure** ASSIGNINGORIGINALIMAGES($\mathbf{M}_{\mathsf{priv}}, n_{\mathsf{priv}}$)
2:            ▷ $\mathbf{M}_{\mathsf{priv}} \in \mathbb{R}^{m \times m}$ is Private Gram matrix, $n_{\mathsf{priv}}$ is the number of private images
3:      $\mathbf{M}_G \leftarrow \mathbf{M}_{\mathsf{priv}} - \mathbf{I}$
4:      **if** $n_{\mathsf{priv}} < 5$ **then**
5:          **for** $H \in \{0,1\}^{n_{\mathsf{priv}} \times n_{\mathsf{priv}}}$ **do**
6:              $\mathbf{M}_H \leftarrow$ adjacency matrix of the line graph of $H$
7:              **if** $\mathbf{M}_H = \mathbf{M}_G$ **then**
8:                  $\widetilde{\mathbf{W}} \leftarrow \widetilde{\mathbf{W}} \cup \{\mathbf{W}_H\}$             ▷ $\mathbf{W}_H$ is the incidence matrix of $H$
9:              **end if**
10:          **end for**
11:          **return** $\widetilde{\mathbf{W}}$
12:      **end if**
13:      Reconstruct $G$ from $\mathbf{M}_G$                                 ▷ By Theorem C.2
14:      **return** $\mathbf{W}$                                         ▷ The incidence matrix of $G$
15: **end procedure**

---

**Definition 3.5** (Distinguishable)**.** *For matrix $\mathbf{M} \in \mathbb{R}^{m \times m}$, we say $\mathbf{M}$ is distinguishable if there exists unique solution $\mathbf{W} = (w_1, \ldots, w_m)^\top$ (up to permutation of rows) to the equation $\mathbf{W}\mathbf{W}^\top = \mathbf{M}$ such that $w_i \in \mathrm{supp}(\mathcal{D}_{\mathsf{priv}})$ for all $i \in [m]$.*

**Lemma 3.6** (Assign InstaHide images to the original images)**.** *When $m = \Omega(n_{\mathsf{priv}} \log n_{\mathsf{priv}})$, let $\mathbf{W}_{\mathsf{priv}} = (w_1, \ldots, w_m)^\top$ where $w_i, i \in [m]$ are sampled from distribution $\mathcal{D}_{\mathsf{priv}}$ and $\mathbf{M}_{\mathsf{priv}} = \mathbf{W}_{\mathsf{priv}}\mathbf{W}_{\mathsf{priv}}^\top \in \mathbb{R}^{m \times m}$. Then with high probability $\mathbf{M}_{\mathsf{priv}}$ is distinguishable and algorithm ASSIGNINGORIGINALIMAGES inputs private Gram matrix $\mathbf{M}_{\mathsf{priv}} \in \{0, 1, 2\}^{m \times m}$ correctly outputs $\mathbf{W}_{\mathsf{priv}} \in \{0, 1\}^{m \times n_{\mathsf{priv}}}$ with row sparsity $k_{\mathsf{priv}} = 2$ such that $\mathbf{W}_{\mathsf{priv}}\mathbf{W}_{\mathsf{priv}}^\top = \mathbf{M}_{\mathsf{priv}}$. Furthermore ASSIGNINGORIGINALIMAGES runs in time $O(mn_{\mathsf{priv}})$.*

The proof of Lemma 3.6 is deferred to Appendix C. We consider graph $G = (V, E), |V| = n_{\mathsf{priv}}$ and $|E| = m$ where each $v_i \in V$ corresponds to an original image in private data set and each $e = (v_i, v_j) \in E$ correspond to an encrypted image generated from two original images corresponding to $v_i$ and $v_j$. We define the Gram matrix of graph $G = (V, E)$, denoted by $\mathbf{M}_G \in \{0, 1, 2\}^{m \times m}$ where $m = |E|$, to be $\mathbf{M}_G = \mathbf{W}\mathbf{W}^\top - \mathbf{I}$ where $\mathbf{W} \in \{0, 1\}^{m \times n_{\mathsf{priv}}}$ is the incidence matrix of G. That is [3]

$$\mathbf{M}_G = \begin{bmatrix} |e_1 \cap e_1| & \cdots & |e_1 \cap e_m| \\ \vdots & \ddots & \vdots \\ |e_m \cap e_1| & \cdots & |e_m \cap e_m| \end{bmatrix} \in \{0, 1, 2\}^{m \times m}.$$

We can see that $\mathbf{M}_G$ actually correspond the line graph $L(G)$ of the graph $G$. We similarly call a graph $G$ distinguishable if there exists no other graph $G'$ such that $G$ and $G'$ have the same Gram matrix (up to permutations of edges), namely $\mathbf{M}_G = \mathbf{M}_{G'}$ (for some ordering of edges). To put it into another word, if we know $\mathbf{M}_G$, we can recover $G$ uniquely. Therefore, recovering $\mathbf{W}$ from $\mathbf{M}$ can be viewed as recovering graph $G$ from its Gram matrix $\mathbf{M}_G \in \mathbb{R}^{m \times m}$, and a graph is distinguishable if and only if its Gram matrix $\mathbf{M}_G$ is distinguishable.

This problem was studied since the 1970s and fully resolved by Whitney Whitney (1992). In fact, from a line graph $L(G)$ one can first identify a tree of the original graph $G$ and then proceed to recover the whole graph. The proof is then completed from well-known facts in random graph theory Erdős & Rényi (1960) that $G$ is connected with high probability when $m = \Omega(n_{\mathsf{priv}} \log n_{\mathsf{priv}})$. This paradigm can potentially be extended to handle $k \geq 3$ case with more information of $G$. Intuitively, this is achievable for a dense subgraph of $G$, such as the local structure identified by Chen et al. (2021). More discussion can be found in Appendix C.

### 3.4 SOLVING A LARGE SYSTEM OF EQUATIONS

In this section, we solve the step 4, recovering all private images by solving an $\ell_2$-regression problem. Formally, given mixup coefficients $\mathbf{W}_{\mathsf{priv}}$ (for private images) and contributions to

---

[3]With high probability, $W$ will not have multi-edge. So, most entries of $M$ will be in $\{0, 1\}$.

InstaHide images from public images $\mathbf{Y}_{\mathsf{pub}}$ we recover all private images $\mathbf{X}_{\mathsf{priv}}$ (up to absolute value).

**Lemma 3.7** (Solve $\ell_2$-regression with hidden signs). *Given* $\mathbf{W}_{\mathsf{priv}} \in \mathbb{R}^{m \times n_{\mathsf{priv}}}$ *and* $\mathbf{Y}_{\mathsf{pub}}, \mathbf{Y} \in \mathbb{R}^{m \times d}$. *For each* $i \in [d]$, *let* $\mathbf{Y}_{*,i} \in \mathbb{R}^m$ *denote the $i$-th column of* $\mathbf{Y}$ *and similarily for* $\mathbf{Y}_{\mathsf{pub}_{*,i}}$, *the following $\ell_2$ regression*

$$\min_{z_i \in \mathbb{R}^{n_{\mathsf{priv}}}} \||\mathbf{W}_{\mathsf{priv}} z_i + \mathbf{Y}_{\mathsf{pub}_{*,i}}| - \mathbf{Y}_{*,i}\|_2.$$

*for all* $i \in [d]$ *can be solve by* SOLVINGSYSTEMOFEQUATIONS *in time* $O(2^m \cdot mn_{\mathsf{priv}}^2 \cdot d)$.

*Proof.* Suppose $\mathbf{W}_{\mathsf{priv}} = [w_1 \quad w_2 \quad \cdots \quad w_m]^\top$. Then, the $\ell_2$-regression we considered actually minimizes

$$\sum_{j=1}^m (|w_j^\top z_i + \mathbf{Y}_{\mathsf{pub}_{j,i}}| - \mathbf{Y}_{j,i})^2 = \sum_{j=1}^m (w_j^\top z_i + \mathbf{Y}_{\mathsf{pub}_{j,i}} - \sigma_j \cdot \mathbf{Y}_{j,i})^2,$$

where $\sigma_j \in \{-1, 1\}$ is the sign of $w_j z_i^*$ for the minimizer $z_i^*$.

Therefore, in Algorithm 3, we enumerate all possible $\sigma \in \{\pm 1\}^m$. Once $\sigma$ is fixed, the optimization problem becomes the usual $\ell_2$-regression, which can be solved in $O(n_{\mathsf{priv}}^\omega + mn_{\mathsf{priv}}^2)$ time. Since we assume $m = \Omega(n_{\mathsf{priv}} \log(n_{\mathsf{priv}}))$ in the previous step, the total time complexity is $O(2^m \cdot mn_{\mathsf{priv}}^2)$.

If $\mathrm{sign}(\mathbf{W}_{\mathsf{priv}} z + \mathbf{Y}_{\mathsf{pub}_{*,i}}) = \sigma$ holds for $z = \min_{z \in \mathbb{R}^{n_{\mathsf{priv}}}} \|\mathbf{W}_{\mathsf{priv}} z + \mathbf{Y}_{\mathsf{pub}_{*,i}} - \sigma \circ \mathbf{Y}_{*,i}\|_2$, then $\sum_{j=1}^m (|w_j^\top z_i + \mathbf{Y}_{\mathsf{pub}_{j,i}}| - \mathbf{Y}_{j,i})^2 = 0$ and $z$ is the unique minimizer of the signed $\ell_2$-regression problem almost surely.

Indeed, if we have for $\sigma \neq \widetilde{\sigma}$, $\mathbf{W}_{\mathsf{priv}}(X_{\mathsf{priv}}^\top)_{*,i} + \mathbf{Y}_{\mathsf{pub}_{*,i}} - \sigma \circ \mathbf{Y}_{*,i} = 0$ and $\mathbf{W}_{\mathsf{priv}} \widetilde{z} + \mathbf{Y}_{\mathsf{pub}_{*,i}} - \widetilde{\sigma} \circ \mathbf{Y}_{*,i} = 0$ hold, then from direct calculations we come to $\mathbf{W}_{\mathsf{priv}} \widetilde{z} = \sigma \circ \widetilde{\sigma} \circ (\mathbf{W}_{\mathsf{priv}}(X_{\mathsf{priv}}^\top)_{*,i} + \mathbf{Y}_{\mathsf{pub}_{*,i}}) - \mathbf{Y}_{\mathsf{pub}_{*,i}}$. This indicates that $\sigma \circ \widetilde{\sigma} \circ (\mathbf{W}_{\mathsf{priv}}(X_{\mathsf{priv}}^\top)_{*,i} + \mathbf{Y}_{\mathsf{pub}_{*,i}}) - \mathbf{Y}_{\mathsf{pub}_{*,i}}$ lies in a $n_{\mathsf{priv}}$-dimensional subspace of $\mathbb{R}^m$. Noting that $(X_{\mathsf{priv}}^\top)_{j,i}$ and $\mathbf{Y}_{\mathsf{pub}_{j,i}}$ are i.i.d sampled from Gaussian, the event above happens with probability zero since $m \gg n_{\mathsf{priv}}$. Thus, we can repeat this process for all $i \in [d]$ and solve all $z_i$'s. $\qquad\square$

We also show that $\ell_2$-regression with hidden signs is in fact a very hard problem. Although empirical methods may bypass this issue by directly applying gradient descent, real world practitioners taking shortcuts would certainly suffer from a lack of apriori theoretical guarantees when facing a large private dataset.

**Theorem 3.8** (Lower bound of $\ell_2$-regression with hidden signs, informal version of Theorem D.4.). *There exists a constant $\epsilon > 0$ such that it is NP-hard to $(1 + \epsilon)$-approximate* $\min_{z \in \mathbb{R}^n} \||Wz| - y\|_2$, *where* $W \in \{0, 1\}^{m \times n}$ *is row 2-sparse and* $y \in \{0, 1\}^m$.

We will reduce the MAX-CUT problem to the $\ell_2$-regression. MAX-CUT is a well-known NP-hard problem Berman & Karpinski (1999). A MAX-CUT instance is a graph $G = (V, E)$ with $n$ vertices and $m$ edges. The goal is to find a subset of vertices $S \subseteq V$ such that the number of edges between $S$ and $V \backslash S$ is maximized, i.e., $\max_{S \subseteq V} |E(S, V \backslash S)|$. We can further assume $G$ is 3-regular, that is, each vertex has degree 3. The full proof is deferred to Appendix D.

### 3.5 PUTTING EVERYTHING TOGETHER

Now we are in the position to prove Theorem 3.1.

*Proof.* By Lemma 3.3 the matrix computed in Line 4 satisfies $\mathbf{M} = \mathbf{W}\mathbf{W}^\top$. By Lemma 3.4, Line 7 correctly computes the indices of all public coordinaes of $w_i, i \in [m]$. Therefore from

$$\mathbf{M} = \mathbf{W}\mathbf{W}^\top = \mathbf{W}_{\mathsf{priv}} \mathbf{W}_{\mathsf{priv}}^\top / k_{\mathsf{priv}} + \mathbf{W}_{\mathsf{pub}} \mathbf{W}_{\mathsf{pub}}^\top / k_{\mathsf{pub}},$$

---

**Algorithm 3** Solving a large system of equations

---
1:  **procedure** SOLVINGSYSTEMOFEQUATIONS($\mathbf{W}_{\mathsf{priv}}, \mathbf{Y}_{\mathsf{pub}}, \mathbf{Y}$)
2:                                     $\triangleright \mathbf{W}_{\mathsf{priv}} \in \mathbb{R}^{m \times n_{\mathsf{priv}}}, \mathbf{Y}_{\mathsf{pub}} \in \mathbb{R}^{m \times d}, \mathbf{Y} \in \mathbb{R}^{m \times d}$
3:      **for** $i = 1 \to d$ **do**
4:          $\widetilde{x}_i \leftarrow \emptyset$
5:          **for** $\sigma \in \{-1, +1\}^m$ **do**
6:              $z \leftarrow \min_{z \in \mathbb{R}^{n_{\mathsf{priv}}}} \|\mathbf{W}_{\mathsf{priv}} z + \mathbf{Y}_{\mathsf{pub}_{*,i}} - \sigma \circ \mathbf{Y}_{*,i}\|_2$
7:              **if** $\mathsf{sign}(\mathbf{W}_{\mathsf{priv}} z + \mathbf{Y}_{\mathsf{pub}_{*,i}}) = \sigma$ **then**
8:                  $\widetilde{x}_i \leftarrow \widetilde{x}_i \cup z$
9:              **end if**
10:         **end for**
11:     **end for**
12:     $\widetilde{X} \leftarrow \{\widetilde{x}_1, \cdots, \widetilde{x}_d\}$
13:     **return** $\widetilde{X}$
14: **end procedure**

---

the Gram matrix computed in Line 10 satisfies $\mathbf{M}_{\mathsf{priv}} = \mathbf{W}_{\mathsf{priv}} \mathbf{W}_{\mathsf{priv}}^\top$.

We can now apply Lemma 3.6 to find the private components of mixup weights. Indeed, the output of Line 12 is exactly

$$\mathbf{W}_{\mathsf{priv}} = k_{\mathsf{priv}} \cdot \begin{bmatrix} \mathsf{priv}(\mathbf{W}_{1,*}) \\ \vdots \\ \mathsf{priv}(\mathbf{W}_{m,*}) \end{bmatrix} \in \{0,1\}^{m \times n_{\mathsf{priv}}}.$$

Based on the correctness of private weights, the output in Line 15 is exactly all private images by Lemma 3.7. This completes the proof of correctness of Algorithm 1.

By Lemma 3.3, Line 4 takes $O(m^2 d)$ time. By Lemma 3.4 Line 7 runs in time $O(dn_{\mathsf{pub}}^2 + n_{\mathsf{pub}}^{2\omega+1})$. By Lemma 3.6 private weights can be computed in time $mn_{\mathsf{priv}}$. Line 10 and Line 14 can be computed efficiently in time $O(m^\omega)$. Finally Line 15 is computes in time $O(2^m \cdot mn_{\mathsf{priv}}^2 \cdot d)$ by Lemma 3.7. Combining all these steps we have that the total running time of Algorithm 1 is bounded by $O(m^2 d + dn_{\mathsf{pub}}^2 + n_{\mathsf{pub}}^{2\omega+1} + mn_{\mathsf{priv}}^2 + 2^m \cdot mn_{\mathsf{priv}}^2 d)$.     $\square$

## 4   CONCLUSION AND FUTURE WORK

We show that $\Omega(n_{\mathsf{priv}} \log n_{\mathsf{priv}})$ samples suffice to recover all private images under the current setup for InstaHide challenge of mixing two private images. We show that a key step in attacking procedure can be formulated as a problem of graph isomorphism and prove the uniqueness and hardness of recovery. Our approach has significantly advanced the state-of-the-art approach Chen et al. (2021) that requires $n_{\mathsf{priv}}^{4/3}$ samples to recover a single private image. In addition, we present a theoretical framework to reason about the similarities and differences of existing attacks Carlini et al. (2020); Chen et al. (2021) and our attack on Instahide.

Based on our framework, there are several interesting directions for future study:

- How to generalize our result to recover all private images when mixing more than two private images?

- How to extend this framework to analyze multi-task phase retrieval problem with real-world data?

Real-world security is not a binary issue. We hope that our theoretical contributions shed light on the discussion of safety for distributed training algorithms and provide inspirations for the development of better practical privacy-preserving machine learning methods.

## 5 ETHICS STATEMENT

Our work can potentially create negative societal impacts because our theoretical results can inspire new efficient practical attack methods to machine learning models protected by InstaHide.

## 6 REPRODUCIBILITY STATEMENT

This is a theory paper. We explicitly stated our assumptions and we provided the complete proofs in supplementary materials.

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

# A A UNIFIED FRAMEWORK TO COMPARE WITH EXISTING ATTACKS

| Refs | Rec. | $k_{\mathsf{priv}}$ | Samples | Step 1 | Step 2 | Step 3 | Step 4 |
|------|------|------|---------|--------|--------|--------|--------|
| Chen | one | $\geq 2$ | $m \geq n^{k_{\mathsf{priv}}-2/(k_{\mathsf{priv}}+1)}$ | $dm^2$ | $dn_{\mathsf{pub}}^2 + n_{\mathsf{pub}}^{2\omega+1}$ | $m^2$ | $2^{k_{\mathsf{priv}}^2}$ |
| Ours | all | $= 2$ | $m \geq n_{\mathsf{priv}} \log n_{\mathsf{priv}}$ | $dm^2$ | $dn_{\mathsf{pub}}^2 + n_{\mathsf{pub}}^{2\omega+1}$ | $mn_{\mathsf{priv}}$ | $2^m \cdot n_{\mathsf{priv}}^2 d$ |

Table 1: A summary of running times in different steps between ours and Chen et al. (2021). This table only compares the theoretical result. Let $k_{\mathsf{priv}}$ denote the number of private images we select in InstaHide image. Let $d$ denote the dimension of image. Let $n_{\mathsf{pub}}$ denote the number of images in public dataset. Let $n_{\mathsf{priv}}$ denote the number of images in private dataset. We provide a computational lower bound for Step 4 in Appendix D. There is no algorithm that solves Step 4 in $2^{o(n_{\mathsf{priv}})}$ time under Exponential Time Hypothesis (ETH) (Theorem D.4). Let **Rec.** denote the Recover.

Our attack algorithm (Algorithm 1) contains four steps for $k_{\mathsf{priv}} = 2$. We can prove $m = O(n_{\mathsf{priv}} \log(n_{\mathsf{priv}}))$ suffices for exact recovery. Our algorithm shares similarities as two recent attack results : one is a practical attack Carlini et al. (2020), the other is a theoretical attack Chen et al. (2021). In the next few paragraphs, we describe our attack algorithm in four major steps. For each step, we also give a comparison with the corresponding step in Carlini et al. (2020) and Chen et al. (2021).

- **Step 1.** Section 3.1. Recover the Gram matrix $\mathbf{M} = \mathbf{WW}^\top \in \mathbb{R}^{m \times m}$ of mixup weights $\mathbf{W}$ from synthetic images $\mathbf{Y}$. This Gram matrix contains all inner products of mixup weights $\langle w_i, w_j \rangle$. Intuitively this measures the similarity of each pair of two synthetic images and is a natural start of all existing attacking algorithms.
    - For this step, Carlini et al. (2020)'s attack uses a pre-trained neural network on public dataset to construct the Gram matrix.
    - For this step, note that $\mathbf{Y}$ follows folded Gaussian distribution whose covariance matrix is directly related to $\mathbf{M}$. We can thus solve this step by estimating the covariance of a folded Gaussian distribution. This is achieved by using Algorithm 2 in Chen et al. (2021). It takes $O(m^2 d)$ time.

- **Step 2.** Section 3.2. Recover all public image coefficients and substract the contribution of public coefficients from Gram matrix $\mathbf{M}$ to obtain $\mathbf{M}_{\mathsf{priv}}$. This step is considered as one of the main computational obstacle for private image recovery.
    - For this step, Carlini et al. (2020)'s attack: 1) they treat public images as noise, 2) they don't need to take care of the public images' labels, since current InstaHide Challenge doesn't provide label for public images.
    - For this step, we invoke a paradigm in sparse phase retrieval via using general SDP solver to approximate the principle components of the Gram matrix of public coefficients. Chen et al. (2021) proved that this method exactly outputs all public coefficients. The time of this step has two parts : 1) formulating the matrix, which takes $dn_{\mathsf{pub}}^2$, 2) solving a SDP with $n_{\mathsf{pub}}^2 \times n_{\mathsf{pub}}^2$ size matrix variable and $O(n_{\mathsf{pub}}^2)$ constraints, which takes $n_{\mathsf{pub}}^{2\omega+1}$ time Jiang et al. (2020); Huang et al. (2021), where $\omega$ is the exponent of matrix multiplication.

- **Step 3.** Section 3.3. Recover private coefficients $\mathbf{W}_{\mathsf{priv}} \in \mathbb{R}^{m \times n_{\mathsf{priv}}}$ from private Gram matrix $\mathbf{M}_{\mathsf{priv}}$ ($\mathbf{M}_{\mathsf{priv}} = \mathbf{W}_{\mathsf{priv}} \mathbf{W}_{\mathsf{priv}}^\top$), this step takes $O(m \cdot n_{\mathsf{priv}}^2)$ time.
    - For this step, Carlini et al. (2020)'s attack uses $K$-means to figure out cliques and then solves a min-cost max flow problem to find the correspondence between InstaHide image and original image (see Appendix B for detailed discussions).
    - For this step Chen et al. (2021) starts by finding a local structure called "floral matrix" in the Gram matrix. They prove the existence of this local structure when $m \geq n_{\mathsf{priv}}^{k_{\mathsf{priv}}-2/(k_{\mathsf{priv}}+1)}$. Then Chen et al. (2021) can recover private coefficients within that local structure using nice combinatorial properties of the "floral matrix".

- For this step, we note the fact that in $k_{\mathsf{priv}} = 2$ case the mixup matrix corresponds to the incident matrix of a graph $G$ and the Gram matrix corresponds to its line graph $L(G)$ (while $k_{\mathsf{priv}} \geq 3$ cases correspond to hypergraphs). We can then leverage results in graph isomorphism theory to recover the all private coefficients. In particular, when $m \geq \Omega(n_{\mathsf{priv}} \log n_{\mathsf{priv}})$ the private coefficients are uniquely identifiable from the Gram matrix.

- **Step 4.** Section 3.4. Solve $d$ independent $\ell_2$-regression problems to find private images $X_{\mathsf{priv}}$. Given $\mathbf{W}_{\mathsf{priv}} \in \mathbb{R}^{m \times n_{\mathsf{priv}}}$ and $\mathbf{Y} \in \mathbb{R}^{m \times d}$. For each $i \in [d]$, let $\mathbf{Y}_{*,i} \in \mathbb{R}^m$ denote the $i$-th column of $\mathbf{Y}$, we need to solve the following $\ell_2$ regression

$$\min_{z \in \mathbb{R}^{n_{\mathsf{priv}}}} \||\mathbf{W}_{\mathsf{priv}} z + \mathbf{Y}_{\mathsf{pub}_{*,i}}| - |\mathbf{Y}_{*,i}|\|_2.$$

The classical $\ell_2$ regression can be solved in an efficient way in both theory and practice. However, here we don't know the random signs and we have to consider all $2^m$ possibilities. In fact we show that solving $\ell_2$ regression with hidden signs is NP-hard.

- For this step, Carlini et al. (2020)'s attack is a heuristic algorithm that uses gradient descent.
- For this step, we enumerate all possibilities of random signs to reduce it to standard $\ell_2$ regressions. Chen et al. (2021)'s attack is doing the exact same thing as us. However, since their goal is just recovering one private image (which means $m = O(k^2)$) they only need to guess $2^{k^2}$ possibilities.

## B  Summary of the Attack by Carlini et al. Carlini et al. (2020)

This section summarizes the result of Carlini et al, which is an attack of InstaHide when $k_{\mathsf{priv}} = 2$. Carlini et al. (2020). We first briefly describe the current version of InstaHide Challenge. Suppose there are $n_{\mathsf{priv}}$ private images, the InstaHide authors Huang et al. (2020b) first choose a parameter $T$, this can be viewed as the number of iterations in the deep learning training process. For each $t \in [T]$, Huang et al. (2020b) draws a random permutation $\pi_t : [n_{\mathsf{priv}}] \to [n_{\mathsf{priv}}]$. Each InstaHide image is constructed from a private image $i$, another private image $\pi_t(i)$ and also some public images. Therefore, there are $T \cdot n_{\mathsf{priv}}$ InstaHide images in total. Here is a trivial observation: each private image shown up in exactly $2T$ InstaHide images (because $k_{\mathsf{priv}} = 2$). The model in Chen et al. (2021) is a different one: each InstaHide image is constructed from two random private images and some random public images. Thus, the observation that each private image appears exactly $2T$ does not hold. In the current version of InstaHide Challenge, the InstaHide authors create the InstaHide labels (a vector that lies in $\mathbb{R}^L$ where the $L$ is the number of classes in image classification task) in a way that the label of an InstaHide image is a linear combination of labels (i.e., one-hot vectors) of the private images and not the public images. This is also a major difference compared with Chen et al. (2021). Note that Carlini et al. (2020) won't be confused about, for the label of an InstaHide image, which coordinates of the label vector are from the private images and which are from the public images.

- **Step 1.** Recover a similarity[4] matrix $\mathbf{M} \in \{0, 1, 2\}^{m \times m}$.
  - Train a deep neural network based on all the public images, and use that neural network to construct the similarity matrix $\mathbf{M}$.
- **Step 2.** Treat public image as noise.
- **Step 3. Clustering.** This step is divided into 3 substeps.

  The first substep uses the similarity matrix $\mathbf{M}$ to construct $T n_{\mathsf{priv}}$ clusters of InstaHide images based on each InstaHide image such that the images inside one cluster shares a common original image.

  The second substep runs $K$-means on these clusters, to group clusters into $n_{\mathsf{priv}}$ groups such that each group corresponds to one original image.

---

[4]In Carlini et al. (2020), they call it similarity matrix, in Chen et al. (2021) they call it Gram matrix. Here, we follow Carlini et al. (2020) for convenience.

The third substep constructs a min-cost flow graph to compute the two original images that each InstaHide image is mixed from.

- **Grow clusters.** Figure 1 illustrates an example of this step. For a subset $S$ of InstaHide images ($S \subset [m]$), we define $\text{INSERT}(S)$ as

$$\text{INSERT}(S) = S \cup \arg \max_{i \in [m]} \sum_{j \in S} \mathbf{M}_{i,j}$$

  For each $i \in [m]$, we compute set $S_i \subset [m]$ where $S_i = \text{INSERT}^{(T/2)}(\{i\})$.

- **Select cluster representatives.** Figure 1 illustrates an example of this step. Define distance between clusters as

$$\text{dist}(i,j) = \frac{|S_i \cap S_j|}{|S_i \cup S_j|}.$$

  Run $k$-means using metric $\text{dist} : [m] \times [m] \to \mathbb{R}$ and $k = n_{\mathsf{priv}}$. Result is $n_{\mathsf{priv}}$ groups $C_1, \ldots, C_{n_{\mathsf{priv}}} \subseteq [m]$. Randomly select a representative $r_i \in C_i$, for each $i \in [n_{\mathsf{priv}}]$.

- **Computing assignments.** Construct a min-cost flow graph as Figure 2, with weight matrix $\widetilde{\mathbf{W}} \in \mathbb{R}^{m \times n_{\mathsf{priv}}}$ defined as follows:

$$\widetilde{\mathbf{W}}_{i,j} = \frac{1}{|S_{r_j}|} \sum_{k \in S_{r_j}} \mathbf{M}_{i,k}.$$

  for $i \in [m], j \in [n_{\mathsf{priv}}]$. After solving the min-cost flow (Figure 3), construct the assignment matrix $\mathbf{W} \in \mathbb{R}^{m \times n_{\mathsf{priv}}}$ such that $\mathbf{W}_{i,j} = 1$ if the edge from $i$ to $j$ has flow, and 0 otherwise.

- **Step 4.** Recover original image. From Step 3, we have the unweighted assignment matrix $\mathbf{W} \in \{0,1\}^{m \times n_{\mathsf{priv}}}$. Before we recover the original image, we need to first recover the weight of mixing, which is represented by the weighted assignment matrix $\mathbf{U} \in \mathbb{R}^{m \times n_{\mathsf{priv}}}$. To recover weight, we first recover the label for each cluster group, and use the recovered label and the mixed label to recover the weight.

  - First, we recover the label for each cluster, for all $i \in [n_{\mathsf{priv}}]$. Let $L$ denote the number of classes in the classification task of InstaHide application. For $j \in [m]$, let $y_j \in \mathbb{R}^L$ be the label of $j$.

$$\text{label}(i) = \bigcap_{j \in [m], \mathbf{W}_{j,i}=1} \text{supp}(y_j) \in [L].$$

    Then, for $i \in [m]$ and $j \in [n_{\mathsf{priv}}]$ such that $\mathbf{W}_{i,j} = 1$, define $\mathbf{U}_{i,j} = y_{i,\text{label}(j)}$ for $|\text{supp}(y_i)| = 2$ and $\mathbf{U}_{i,j} = y_{i,\text{label}(j)}/2$ for $|\text{supp}(y_i)| = 1$.
    Here, $\mathbf{W} \in \{0,1\}^{m \times n_{\mathsf{priv}}}$ is the unweighted assignment matrix and $\mathbf{U} \in \mathbb{R}^{m \times n_{\mathsf{priv}}}$ is the weighted assignment matrix. For $\mathbf{W}_{i,j} = 0$, let $\mathbf{U}_{i,j} = 0$.

  - Second, for each pixel $i \in [d]$, we run gradient descent to find the original images. Let $\mathbf{Y} \in \mathbb{R}^{m \times d}$ be the matrix of all InstaHide images, $\mathbf{Y}_{*,i}$ denote the $i$-th column of $\mathbf{Y}$.[5]

$$\min_{z \in \mathbb{R}^{n_{\mathsf{priv}}}} \| |\mathbf{U}z| - |\mathbf{Y}_{*,i}| \|_2.$$

## C  MISSING PROOFS FOR THEOREM 3.6

For simplicity, let $\mathbf{W}$ denote $\mathbf{W}_{\mathsf{priv}}$ and $\mathbf{M}$ denote $\mathbf{M}_{\mathsf{priv}}$ in this section.

---

[5]The description of the attack in Carlini et al. (2020) recovers original images by using gradient descent for $\min_{z \in \mathbb{R}^{n_{\mathsf{priv}}}} \|U|z| - |\mathbf{Y}_{*,i}|\|_2$, which we believe is a typo.

## C.1 A GRAPH PROBLEM ($k_{\mathsf{priv}} = 2$)

**Theorem C.1** (Whitney (1992)). *Suppose $G$ and $H$ are connected simple graphs and $L(G) \cong L(H)$. If $G$ and $H$ are not $K_3$ and $K_{1,3}$, then $G \cong H$. Furthermore, if $|V(G)| \geq 5$, then an isomorphism of $L(G)$ uniquely determines an isomorphism of $G$.*

In other words, this theorem claims that given $\mathbf{M} = \mathbf{W}\mathbf{W}^{\top}$, if the underlying $\mathbf{W}$ is not the incident matrix of $K_3$ or $K_{1,3}$, $\mathbf{W}$ can be uniquely identified up to permutation. Theorem C.1 can also be generalized to the case when $G$ has multi-edges Zverovich (1997).

On the other hand, a series of work Roussopoulos (1973); Lehot (1974); Syslo (1982); Degiorgi & Simon (1995); Liu et al. (2015) showed how to efficiently reconstruct the original graph from its line graph:

**Theorem C.2** (Liu et al. (2015)). *Given a graph $L$ with $m$ vertices and $t$ edges, there exists an algorithm that runs in time $O(m + t)$ to decide whether $L$ is a line graph and output the original graph $G$. Furthermore, if $L$ is promised to be the line graph of $G$, then there exists an algorithm that outputs $G$ in time $O(m)$.*

With Theorem C.1 and Theorem C.2, Theorem 3.6 follows immediately:

*Proof of Theorem 3.6.* First, since $m = \Omega(n_{\mathsf{priv}} \log(n_{\mathsf{priv}}))$, a well-known fact in random graph theory by Erdős and Rényi Erdős & Rényi (1960) showed that the graph $G$ with incidence matrix $\mathbf{W}$ will almost surely be connected. Then, we compute $\mathbf{M}_G = \mathbf{M} - \mathbf{I}$, the adjacency matrix of the line graph $L(G)$. Theorem C.1 implies that $G$ can be uniquely recovered from $\mathbf{M}_G$ as long as $n_{\mathsf{priv}}$ is large enough. Finally, We can reconstruct $G$ from $\mathbf{M}_G$ by Theorem C.2.

For the time complexity of Algorithm 2, the reconstruction step can be done in $O(m)$ time. Since we need to output the matrix $\mathbf{W}$, we will take $O(mn_{\mathsf{priv}})$-time to construct the adjacency matrix of $G$. Here, we do not count the time for reading the whole matrix $\mathbf{M}$ into memory.

$\square$

## C.2 GENERAL CASE ($k_{\mathsf{priv}} > 2$)

The characterization of $\mathbf{M}$ and $\mathbf{W}$ as the line graph and incidence graph can be generalized to $k_{\mathsf{priv}} > 2$ case, which corresponds to hypergraphs.

Suppose $\mathbf{M} = \mathbf{W}\mathbf{W}^{\top}$ with $k_{\mathsf{priv}} = k > 2$. Then, $\mathbf{W}$ can be recognized as the incidence matrix of a $k$-uniform hypergraph $G$, i.e., each hyperedge contains $k$ vertices. $\mathbf{M}_G = \mathbf{M} - \mathbf{I}$ corresponds to adjacency matrix of the line graph of hypergraph $G$: $(\mathbf{M}_G)_{i,j} = |e_i \cap e_j|$ for $e_i, e_j$ being two hyperedges. Now, we can see that each entry of $\mathbf{M}_G$ is in $\{0, \ldots, k\}$.

Unfortunately, the identification problem becomes very complicated for hypergraphs. Lovász Lovász (1977) stated the problem of characterizing the line graphs of 3-uniform hypergraphs and noted that Whitney's isomorphism theorem (Theorem C.1) cannot be generalized to hypergraphs. Hence, we may not be able to uniquely determine the underlying hypergraph and we should just consider a more basic problem:

**Problem C.3** (Line graph recognition for hypergraph). *Given a simple graph $L = (V, E)$ and $k \in \mathbb{N}$, decide if $L$ is the line graph of a $k$-uniform hypergraph $G$.*

Even for the recognition problem, it was proved to be NP-complete for fixed $k \geq 3$ Levin & Tyshkevich (1993); Poljak et al. (1981). However, Problem C.3 becomes tractable if we add more constraints to the underlying hypergraph $G$. First, suppose $G$ is a linear hypergraph, i.e., the intersection of two hyperedges is at most one. If we further assume the minimum degree of $G$ is at least 10, i.e., each vertex are in at least 10 hyperedges, there exists a polynomial-time algorithm for the decision problem. Similar result also holds for $k > 3$ Jacobson et al. (1997). Let the edge-degree of a hyperedge be the number of triangles in the hypergraph containing that hyperedge. Jacobson et al. (1997) showed that assuming the minimum edge-degree of $G$ is at least $2k^2 - 3k + 1$, there exists a polynomial-time algorithm

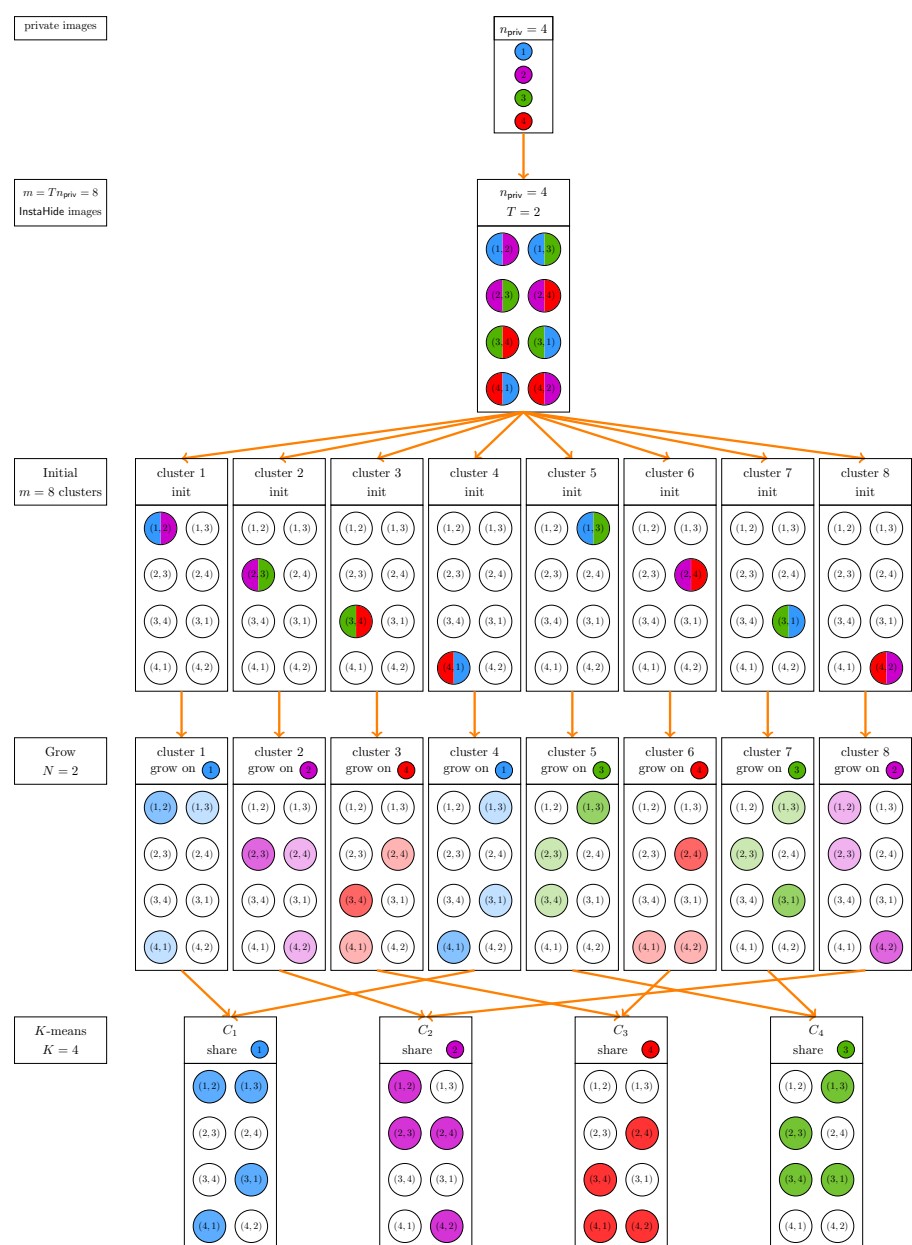

Figure 1: An example about cluster step in Carlini et al. (2020) for $T = 2$ and $n_{\mathsf{priv}} = 4$. First, starting from each InstaHide image (top), the algorithm grows cluster $S_i$ with size 3 (middle). Then, we use $K$-means for $K = 4$ to compute 4 groups $C_1, \ldots, C_4$ (bottom), these groups each correspond to one original image.

to decide whether $L$ is the line graph of a linear $k$-uniform hypergraph. Furthermore, in the yes case, the algorithm can also reconstruct the underlying hypergraph. We also note that without any constraint on minimum degree or edge-degree, the complexity of recognizing line graphs of $k$-uniform linear hypergraphs is still unknown.

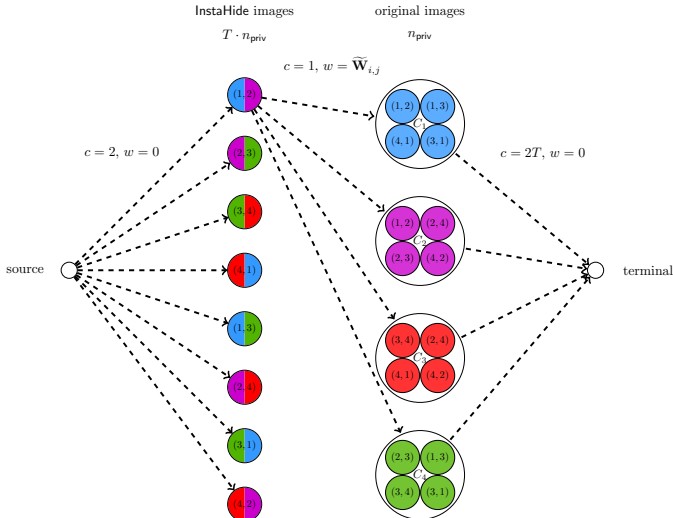

Figure 2: The construction of the graph for min-cost max flow. $c$ denotes the flow capacity of the edge, and $w$ denote the weight of the edge. The graph contains $T \cdot n_{\mathsf{priv}}$ nodes for each InstaHide images, $n_{\mathsf{priv}}$ nodes for each original images, a source and a terminal. There are three types of edges: i) (left) from the source to each InstaHide image node, with flow capacity 2 and weight 0; ii) (middle) from each InstaHide image node $i$ to each original image node $j$, with flow capacity 1 and weight $\widetilde{\mathbf{W}}_{i,j}$; iii) (right) from each original image node to the terminal, with flow capacity $2T$ and weight 0.

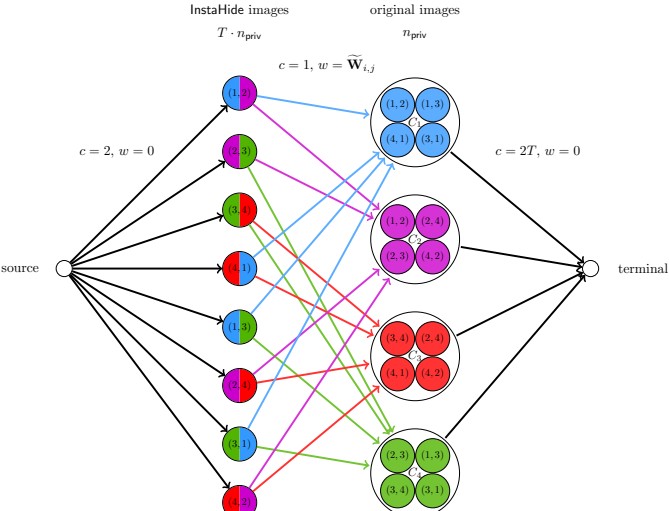

Figure 3: The result of solving the min-cost flow in Figure 2. Each InstaHide image is assigned to two clusters, which ideally correspond to two original images. In reality, a cluster may not contain all InstaHide images that share the same original image.

## D   COMPUTATIONAL LOWER BOUND

The goal of this section is to prove that the $\ell_2$-regression with hidden signs is actually a very hard problem, even for approximation (Theorem D.4), which implies that Algorithm 3 cannot be significantly improved. For simplicity we consider $S_{\mathsf{pub}} = \emptyset$.

We first state an NP-hardness of approximation result for 3-regular MAX-CUT.

**Theorem D.1** (Imapproximability of 3-regular MAX-CUT, Berman & Karpinski (1999))**.** *For every $\epsilon > 0$, it is NP-hard to approximate 3-regular MAX-CUT within a factor of $r + \epsilon$, where $r \approx 0.997$.*

If we assume the Exponential Time Hypothesis (ETH), which a plausible assumption in theoretical computer science, we can get stronger lower bound for MAX-CUT.

**Definition D.2** (Exponential Time Hypothesis (ETH), Impagliazzo & Paturi (2001)). *There exists a constant $\epsilon > 0$ such that the time complexity of n-variable 3SAT is at least $2^{\epsilon n}$.*

**Theorem D.3** (Fotakis et al. (2016)). *Assuming ETH, there exists a constant $0 < r' < 1$ such that no $2^{o(n)}$-time algorithm can $r'$-approximate the MaxCut of an n-vertex, 5-regular graph.*

With Theorem D.1 and Theorem D.3, we can prove the following inapproximability result for the $\ell_2$-regression problem with hidden signs.

**Theorem D.4** (Lower bound of $\ell_2$-regression with hidden signs). *There exists a constant $\epsilon > 0$ such that it is NP-hard to $(1 + \epsilon)$-approximate*

$$\min_{z \in \mathbb{R}^n} \||Wz| - y\|_2, \tag{1}$$

*where $W \in \{0,1\}^{m \times n}$ is row 2-sparse and $y \in \{0,1\}^m$.*

*Furthermore, assuming ETH, there exists a constant $\epsilon'$ such that no $2^{o(n)}$-time algorithm can $\epsilon'$-approximate Eq. (1).*

*Proof.* Given a 3-regular MAX-CUT instance $G$, we construct an $\ell_2$-regression instance $(W, y)$ with $W \in \{0,1\}^{m' \times n}$ and $y \in \{0,1\}^{m'}$ where $m' = m + cn = (1 + 3c/2)m$ and $c = 10^6$ as follows.

- For each $i \in [m]$, let the $i$-th edge of $G$ be $e_i = \{u, v\}$. We set $W_{i,*}$ to be all zeros except the $u$-th and $v$-th coordinates being one. That is, we add a constraint $|z_u + z_v|$. And we set $y_i = 0$.

- For each $j \in [n]$, we set $W_{m+c(j-1)+1,*}, \ldots, W_{m+cj,*}$ to be all zero vectors except the $j$-th entry being one. That is, we add $c$ constraints of the form $|z_j|$. And $y_{m+c(j-1)+1} = \cdots = y_{m+cj} = 1$.

**Completeness.** Let opt be the optimal value of max-cut of $G$ and let $S_{\text{opt}}$ be the optimal subset. Then, for each $u \in S_{\text{opt}}$, we set $z_u = 1$; and for $u \notin S_{\text{opt}}$, we set $z_u = -1$. For the first type constraints $|z_u + z_v|$, if $u$ and $v$ are cut by $S_{\text{opt}}$, then $|z_u + z_v| = 0$; otherwise $|z_u + z_v| = 2$. For the second type constraints $|z_j|$, all of them are satisfied by our assignment. Thus, $\|Wz - y\|_2^2 = 4(m - \text{opt})$.

**Soundness.** Let $\eta$ be a constant such that $r < \eta < 1$, where $r$ is the approximation lower bound in Theorem D.1. Let $\delta = \frac{1-\eta}{10c}$. We will show that, if there exits a $z$ such that $\|Wz - y\|_2^2 \le \delta m'$, then we can recover a subset $S$ with cut-size $\eta m$.

It is easy to see that the optimal solution lies in $[-1,1]^n$. Since for $z \notin [-1,1]^n$, we can always transform it to a new vector $z' \in [-1,1]^n$ such that $\|Wz' - y\|_2 \le \|Wz - y\|_2$.

Suppose $z \in \{-1,1\}^n$ is a Boolean vector. Then, we can pick $S = \{i \in [n] : z_i = 1\}$. We have the cut-size of $S$ is

$$
\begin{aligned}
|E(S, V \setminus S)| &\ge m - \delta m'/4 \\
&= m - \delta(1 + 3c/2)m/4 \\
&= (1 - \delta/4 - 3c\delta/8)m \\
&\ge \eta m,
\end{aligned}
$$

where the last step follows from $\delta \le \frac{8(1-\eta)}{2+6c}$.

For general $z \in [-1,1]^n$, we first round $z$ by its sign: let $\overline{z}_i = \text{sign}(z_i)$ for $i \in [n]$. We will show that

$$\|W\overline{z} - y\|_2^2 - \|Wz - y\|_2^2 \le \frac{48}{c}m$$

which implies

$$\|W\bar{z} - y\|_2^2 = \|Wz - y\|_2^2 + (\|W\bar{z} - y\|_2^2 - \|Wz - y\|_2^2)$$
$$\leq \delta m' + \frac{48}{c} m.$$

Then, we have the cut-size of $S$ is

$$|E(S, V\backslash S)| \geq m - (\delta m' - 48m/c)/4$$
$$= (1 - \delta/4 - 3c\delta/8 - 12/c)m$$
$$\geq \eta m,$$

where the last step follows from $\delta \leq \frac{8(1-\eta-12/c)}{2+6c}$.

Let $\Delta_i := |\bar{z}_i - z_i| = 1 - |z_i| \in [0, 1]$. We have

$$\|W\bar{z} - y\|_2^2 - \|Wz - y\|_2^2 = \sum_{i=1}^{m} (\bar{z}_{u_i} + \bar{z}_{v_i})^2 - (z_{u_i} + z_{v_i})^2 + c \cdot \sum_{j=1}^{n} (|\bar{z}_j| - 1)^2 - (|z_j| - 1)^2$$
$$= \sum_{i=1}^{m} (\bar{z}_{u_i} + \bar{z}_{v_i})^2 - (z_{u_i} + z_{v_i})^2 - c \cdot \sum_{j=1}^{n} (|z_j| - 1)^2$$
$$= \sum_{i=1}^{m} (\bar{z}_{u_i} + \bar{z}_{v_i})^2 - (z_{u_i} + z_{v_i})^2 - c \cdot \sum_{j=1}^{n} \Delta_j^2$$
$$\leq \sum_{i=1}^{m} 4|\Delta_{u_i} + \Delta_{u_j}| - c \cdot \sum_{j=1}^{n} \Delta_j^2$$
$$= \sum_{i=1}^{n} 12\Delta_i - c\Delta_i^2$$
$$\leq \frac{72}{c} n$$
$$= \frac{48}{c} m,$$

where the first step follows by the construction of $W$ and $y$. The second step follows from $|\bar{z}_j| = 1$ for all $j \in [n]$. The third step follows from the definition of $\Delta_j$. The forth step follows from $|z_{u_i} + z_{u_j}| \in [0, 2]$. The fifth step follows from the degree of the graph is 3. The fifth step follows from the minimum of the quadratic function $12x - cx^2$ in $[0, 1]$ is $\frac{72}{c}$. The last step follows from $m = 3n/2$.

Therefore, by the completeness and soundness of reduction, if we take $\epsilon \in (0, \delta)$, Theorem D.1 implies that it is NP-hard to $(1 + \epsilon)$-approximate the $\ell_2$-regression, which completes the proof of the first part of the theorem.

For the furthermore part, we can use the same reduction for a 5-regular graph. By choosing proper parameters ($c$ and $\delta$), we can use Theorem D.3 to rule out $2^{o(n)}$-time algorithm for $O(1)$-factor approximation. We omit the details since they are almost the same as the first part. □

