# OpenReview forum: "InstaHide’s Sample Complexity When Mixing Two Private Images "
_ICLR.cc/2022/Conference — ICLR 2022 Submitted_

### Official Review · Reviewer_x8b2 · 2021-10-24

**Correctness:** 4
**Technical Novelty And Significance:** 3
**Empirical Novelty And Significance:** 3
**Recommendation:** 6
**Confidence:** 4

**Details Of Ethics Concerns:**

The authors adequately addressed the limitations and potential negative societal impact of their work.

**Main Review:**

Inspired by previous works (e.g., Chen, Song, and Zhuo’20), the paper assumes that the images were sampled from a Gaussian distribution and the number of images needed to be mixed via InstaHide is 2 (i.e., $k=2$). Assumptions of this sort in this context are somewhat acceptable and used in practice.

This paper is a generalization of the paper by Chen et al. (2021), where the sample size has been exponentially improved, while the time needed to find such a sample worsened. The time in the paper was shown to be exponential in the number of the hidden images, due to a reduction to solving an instance of a regression problem where the signs of the entries are unknown or hidden. Although the time needed to design the attack is exponential in the number of private images, the theoretical guarantees are favorable.

Although the theory seems sound, however, the weakness of this paper lies in its practicability. Is it possible to apply some data reduction techniques (e.g., coresets) of the data with respect to your variant of the $\ell_2$-regression? This would help in reducing the time needed to solve the problem. However, note that using such technology would result in an approximated solution to the regression problem.

**Summary Of The Paper:**

The paper studies the weaknesses of InstaHide and provides an algorithm for retrieving all private images using near-linear size samples from the distribution of the private images.

**Summary Of The Review:**

First, the paper is well written and the theoretical results are sound.  Secondly, the paper indeed sheds light on the weaknesses of InstaHide. The authors have in addition shown the hardness of the involved regression problem with hidden signs, which is responsible for a large amount of time (theoretically speaking) needed for the suggested attack on InstaHide.

---

> ### Author Response · Authors · 2021-11-23
> **Thank you for your valuable feedback.**
>
> Thank you for your valuable feedback. We address the questions and comments in the following. We hope given these clarifications you will consider increasing your score.
>
> - "Is it possible to apply some data reduction techniques (e.g., coresets) of the data with respect to your variant of the ℓ2-regression?"
>
> We thank the reviewer for raising this interesting further direction. We note that our lower bound only holds when the private images are chosen adversarially. On average, this problem could be solved in polynomial time. For example, our method can easily recover the heavy pixels of all the private images. Here, “heavy” means it is $k_{priv}$ times larger in absolute value than the average pixel value in that location across all private images. Since we know that the private images are chosen uniformly at random, we can consider an estimator $\hat z = E_S [ \langle 1_S, z\rangle^2 \cdot  1_S]$ over random subset $S$, which is the expectation of the entry-wise square of the vector $Wz$ when $z$ is fixed and $W$ is a random matrix. Then, we can prove that for heavy coordinate $i$, $\hat z_i$ is close to $z_i^2 + E_S[ \langle 1_S,z \rangle^2] \cdot (k_{priv} / n_{priv})$. Hence, we can form an unbiased estimator for $E_S[\langle 1_S,z \rangle^2]$ via $\frac{1}{m}\sum_{i=1}^m w_i \cdot y_i^2$, where $w_i$ is the $i$-th column of $W$. Therefore, in average-case, we can recover the “heavy” pixels in time $O(m \cdot n_{priv})$.

---

### Official Review · Reviewer_3gkW · 2021-11-03

**Correctness:** 4
**Technical Novelty And Significance:** 2
**Empirical Novelty And Significance:** Not applicable
**Recommendation:** 5
**Confidence:** 3

**Main Review:**

As a theoretical paper, I consider the results solid and complete (I did not checked all the details of the proof). Although it brings the bound to nearly optimal for a particular settings (k_priv = 2), but it seems to be a common settings and great starting point.

My main concern of this submission is its limited scope: the conclusion provides limited value for people outside of this area. It is unlikely to benefit practitioners or readers who works on other theoretical problems other than InstaHide. However, due to the significance of the privacy related topics, I would err on the side of accepting.

In order to provide more value to a larger audience: It will be great if the authors can provide more discussion for cases where k_priv > 2, it will help readers who are not familiar with the InstaHide setting (like myself) to gain more insight on the problem and it will have better utilities for the field as a whole. Additional assumptions or empirical demonstration would also help.

**Summary Of The Paper:**

In this paper, the authors examines the sample complexity for recovery private images in the InstaHide setting. Comparing with existing works, the authors provided a tighter sample complexity bound when k_{priv} =2, which is nearly-optimal up to logarithmic factors at the cost of an increased time-complexity.

**Summary Of The Review:**

The authors provides a solid solution for a problem with limited scope. In the current form, I believe the value for the general readers are limited, therefore I would recommend for weak reject. It will be great if the authors can generalize the results for a larger impact.

---

> ### Author Response · Authors · 2021-11-23
> **Thank you for your valuable feedback.**
>
> Thank you for your valuable feedback. We address the questions and comments in the following. We hope given these clarifications you will consider increasing your score.
>
> - "My main concern of this submission is its limited scope: the conclusion provides limited value for people outside of this area. "
>
> We would like to comment about generalizing the presented algorithm to other data mix ways ([1,2]). Our work can be seen as a solution to the “batched k-sum” problem introduced in the Instahide paper. Previously, there were only theoretical results for the classical $k$-SUM problem.  Therefore, our general attack framework deals with any data mixing method with “batched k-sum” structure.
>
> - "In order to provide more value to a larger audience: It will be great if the authors can provide more discussion for cases where k_priv > 2"
>
> For k_priv > 2, the problem becomes recovering a hypergraph from its intersection graph. There are no efficient algorithms for recognizing if a Gram matrix is the intersection graph of a hypergraph. In some special cases, for example, when every pair of distinct vertices are adjacent in at most one edge, the problem of recovering hypergraphs from line intersection graphs might be easier. These types of hypergraphs are called “linear hypergraphs” and several works shows that they can be recognized and recovered in the condition that their minimum edge degree is lower bounded by a polynomial term of k_priv.
>
>
> However, one prominent issue is if hypergraph random generated by Mixup or InstaHide satisfies these conditions. In fact, with high probability hypergraph random generated by Mixup or InstaHide must have at least one pair of distinct vertices that are adjacent in two edges. Hence the hypergraph is not linear. Therefore it is challenging to use current graph theories in the setup of Instahide.
>
> Further, the general problem of boolean matrix factorization M = W^\top W can be efficiently solved in the average-case [3]. The average-case recovery means that boolean matrix factorization can be resolved with high probability to the randomness of W. Therefore this result can be used in our setting to achieve the corresponding guarantee. Revealing more structural conditions where worst-case recovery is possible and developing efficient algorithms for this task seems a challenging and interesting further direction.
>
>
> [1]. Yangsibo Huang et al. TextHide: Tackling Data Privacy in Language Understanding Tasks. 2020.
>
> [2]. Eitan Borgnia et al. DP-InstaHide: Provably Defusing Poisoning and Backdoor Attacks with Differentially Private Data Augmentations. 2021.
>
> [3] Sitan Chen et al. Symmetric Sparse Boolean Matrix Factorization and Applications. 2022.

---

### Official Review · Reviewer_d4iK · 2021-11-06

**Correctness:** 4
**Technical Novelty And Significance:** 2
**Empirical Novelty And Significance:** Not applicable
**Recommendation:** 5
**Confidence:** 4

**Main Review:**


Techniques:

The overall framework used in the paper is very similar to previous attacks on InstaHide by Carilini  et al [CDG+20] and Chen et al [CSZ20]. From my understanding, the main innovation seems to be in finding W such that WW^T = M, where M is the gram matrix. Here the paper deviates from the previous approaches, treats the problem as a question in Graph theory, and uses a result due to Whitney. This connection is simple but neat. Rest of the attack is similar to CSZ20 as I understand.


**Summary Of The Paper:**

The paper studies the sample complexity of InstaHide challenge in an idealized setting, where it is assumed that each public and private image is generated using a standard Gaussian distribution. Further, similar to previous attacks, the paper assumes that each image is a combination of 2 private images.

The main contribution of the paper is to show that using O(n polylog ) images one can recover all private images in the InstaHide challenge. This is an improvement over previous works in two ways: 1) The paper shows how to recover all images rather than few (or 1) images. 2) The previous best sample complexity for this case was n^4/3. However, on the downside, the running time of the algorithm is exponential in the number of private images.

**Summary Of The Review:**

The paper makes reasonable improvements to the sample complexity of InstaHide challenge. However, I have two main concerns:

1) Is the attack in the paper gives any significant improvement in real world application? I feel that it is not, and paper has no experiments. Given that InstaHide is rather new suggestion for privacy in ML and has not really been widely used, so I am not sure if this is a significant improvement in practice. I would like to know authors take on this.

2) My understand that finding W such that WW^T = M by applying Whitney's theorem seems to be the only difference compared to CSZ20 paper. Is this correct? It would be worthwhile to summarize technical improvements over CSZ20.

---

> ### Author Response · Authors · 2021-11-23
> **Thank you for your valuable feedback.**
>
> Thank you for your valuable feedback. We address the questions and comments in the following. We hope given these clarifications you will consider increasing your score.
>
> - "Is the attack in the paper gives any significant improvement in real world application?"
>
> The attack algorithm in the paper aims at addressing the theoretical question of recovering Gaussian images from InstaHide encoding. We note two practical implications that might be of interest in real-world applications. First, the unifying framework in Appendix partially explains how the practical attack by Carlini et al. (2020) works. Second, the lower bound in section D reveals that InstaHide is computationally sound. This confirms the conjecture in Appendix C of InstaHide paper, where the authors relate the batched k-sum problem (defined in Problem 1) to the classical k-sum and conjecture that instance encoding enjoys the similar hardness therein.
>
> - "My understand that finding W such that WW^T = M by applying Whitney's theorem seems to be the only difference compared to CSZ20 paper. Is this correct?"
>
> Yes, technically the algorithm improves over CSZ20 by the third step (recover private coefficients from private Gram matrix). The major technique is to reduce the problem to a graph problem of recovering a graph randomly generated by mixup from its line graph. We use a key observation that as long as the number of samples is greater than $n \log n$, the graph is connected with high probability, and therefore can be identified via the line graph.

---

> > ### Comment · Reviewer_d4iK · 2021-11-29
> > **Thank you**
> >
> > Thank you for the your explanations.

---

### Decision · Program_Chairs · 2022-01-20

**Decision:**

Reject

**Comment:**

All reviewers agree that this is a reasonable contribution but that it is also extremely limited in scope. The authors suggest in one of their response that their technique could apply to "any data mixing method with “batched k-sum” structure". Such a larger level of generality might make the paper more interesting, but at the moment it is an extremely niche result.